# How to Get Spiking LLMs? A Dual ANN-to-SNN Conversion with Layer-Wise Calibration

## Abstract

With rising concerns about data privacy, deploying large language models (LLMs) on edge devices rather than relying solely on cloud-based solutions is becoming increasingly essential. Nonetheless, the constrained power and computational capacity of edge hardware frequently hinder the practical deployment of LLMs. Spiking Neural Networks (SNNs) have gained attention as a viable alternative, offering brain-inspired efficiency and low power consumption, making them ideal for edge deployment. Among various SNN training strategies, ANN-to-SNN conversion stands out for its relatively low computational cost compared to training spiking networks from scratch. However, conventional conversion methods still require a specially trained, conversion-friendly ANN, which becomes prohibitively expensive when applied to large-scale models like LLMs. To address this limitation, we propose a novel ANN-to-SNN conversion framework that can be regarded as a dual version of conventional conversion methods. Built on quantized LLMs, our approach eliminates the need to train a dedicated ANN tailored for conversion. A key challenge in such conversions is the temporal dynamics of spike arrivals—commonly known as unevenness error—which can cause significant performance degradation. To mitigate this issue, we introduce a parameter-efficient, layer-wise calibration technique that effectively reduces conversion errors, particularly unevenness error, while keeping computational overhead minimal. Theoretical analysis demonstrates that our calibration method substantially lowers the final conversion error between the original LLM and its spiking counterpart. Extensive experiments on LLaMA models show that our method achieves performance comparable to state-of-the-art quantization techniques, highlighting its effectiveness.

## 1 Introduction

Large language models (LLMs) are a type of AI model trained on vast amounts of text data to learn the patterns and structures of language, enabling them to perform a wide range of language-related tasks Lindemann (2023); Bevilacqua et al. (2025); Liu et al. (2023a); Jia et al. (2024). In recent years, their capabilities have grown dramatically, with model sizes scaling from approximately one billion to over one trillion parameters. The rapid growth in parameter size necessitates immense computational power and energy, which restricts the deployment of LLMs primarily to cloud platforms. However, reliance on the cloud introduces significant challenges, including increased network latency and heightened risks of data privacy breaches. As a result, there is a growing interest in alternative neural architectures that can be deployed on edge devices while maintaining comparable performance with significantly reduced energy consumption.

One promising direction for enabling energy-efficient LLMs on edge devices is the integration of spiking neural networks (SNNs), which are biologically inspired by the event-driven communication of neurons. Current methods for obtaining SNNs can be broadly categorized into two paradigms: direct training (DT) and ANN-to-SNN conversion. Direct training involves optimizing spiking models from scratch using backpropagation with surrogate gradients, as demonstrated in models like SpikeGPT Zhu et al. (2023) and SpikeBERT Lv et al. (2023). While DT has shown promise, it remains computationally intensive and faces significant challenges related to gradient estimation, thereby limiting its applicability to small-scale models. In contrast, the ANN-to-SNN conversion approach transfers weights from a pre-trained neural network to its spiking counterpart Cao et al. (2015). This strategy preserves the performance of the original model while substantially reducing

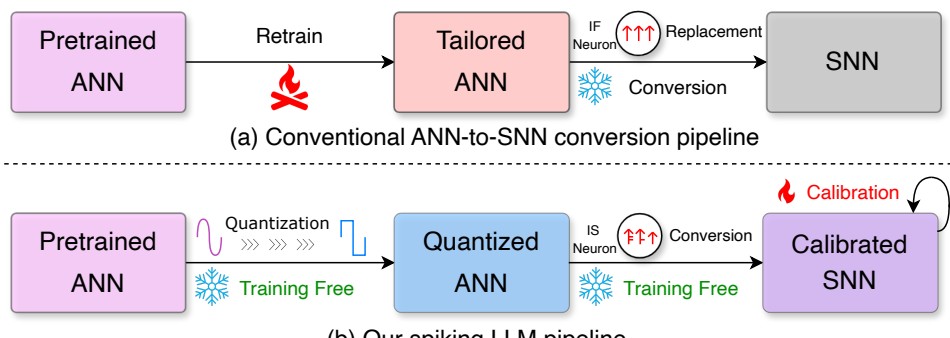

Figure 1: The difference of conventional ANN-to-SNN conversion and our spiking LLM pipeline.

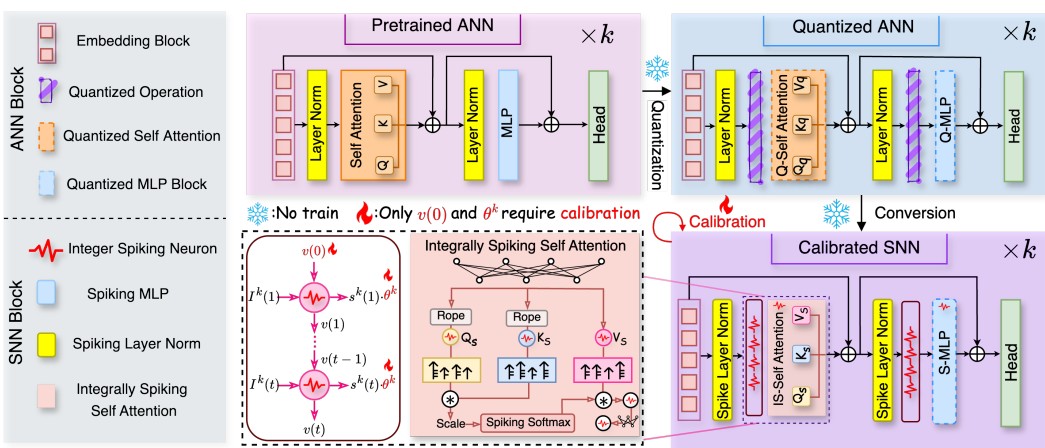

Figure 2: Our complete dual conversion is divided into ANN-to-SNN and SNN calibration. The former obtains SNN with significant errors, and the latter greatly reduces the error of SNN.

training costs. By leveraging the robustness of existing LLMs, ANN-to-SNN conversion enables scalable and efficient deployment of spiking models. Consequently, it has emerged as the dominant approach in this domain, exemplified by recent advances such as SpikeZIP You et al. (2024).

However, applying existing ANN-to-SNN conversion methods to obtain spiking LLMs remains highly challenging. These methods typically depend on specially trained, conversion-friendly ANNs as shown in Figure 1 (a), which become prohibitively expensive to scale when applied to large models such as LLMs. Furthermore, conventional spiking neurons—such as the Integrate-and-Fire (IF) model Diehl et al. (2015); Sengupta et al. (2019); Bu et al. (2022)—exhibit limited expressive capacity, making it difficult to faithfully reproduce the continuous activation patterns of ANNs. A further complication arises from the unevenness error Bu et al. (2022), which results from temporal inconsistencies in spike sequences arriving at downstream layers. These timing discrepancies introduce significant deviations in neural activations, particularly in high time-step settings where the mismatch between spiking dynamics and ANN behavior becomes more severe. Collectively, these limitations not only impair the accuracy of the converted models but also undermine their scalability and practical deployment in real-world applications.

To address the aforementioned limitations, we propose a novel dual ANN-to-SNN conversion framework that eliminates the need for training a specialized, conversion-friendly ANN, as shown in Figure 1 (b). Our approach begins with a statically quantized LLM and introduces an integer-based spiking neuron equipped with multi-hierarchical thresholds, replacing conventional quantization functions. This design significantly enhances the expressive power of spiking neurons, enabling a more accurate representation of the original ANN activations and facilitating the effective conversion of LLMs into their spiking counterparts. To further reduce the mismatch between the target LLM and the resulting SNN, we develop a parameter-efficient, layer-wise calibration strategy that minimizes

conversion error, particularly unevenness error, without incurring substantial computational overhead. The theoretical analysis shows that this calibration method significantly reduces the final conversion error between the original LLM and its spiking counterpart. Our framework is depicted in Figure 1(b). Extensive experiments on LLaMA models show that our method achieves performance comparable to state-of-the-art quantization techniques, highlighting its effectiveness.

The contributions of this paper are summarized as follows:

1. We introduce a novel dual ANN-to-SNN conversion framework that removes the requirement for training a conversion-specific ANN, enabling more scalable to large language models.

2. We develop a theory-backed layer-wise calibration method that achieves significant conversion error reduction with minimal computational and memory overhead.

3. Most importantly, this work serves as a seed effort toward building a spiking LLM that achieves comparable performance with well-established large models while maintaining the neural dynamics of SNNs, and potentially reduces the energy consumption of LLMs.

## 2 RELATED WORK

### 2.1 QUANTIZED LLMS

Quantization Shao et al. (2024) has become a promising method to reduce model size and memory usage for efficient deployment on resource-limited devices. This research can be divided into two main categories: quantization-aware training (QAT) Liu et al. (2023b) and post-training quantization (PTQ) Xiao et al. (2023). QAT trains quantized model weights with extra data using techniques like straight-through estimator Chen et al. (2024b); Du et al. (2024), but is costly for LLMs. Conversely, PTQ, favored for LLMs, employs minimal calibration data and dynamic activation quantization to address accuracy issues arising from activation outliers during inference Frantar et al. (2023). Techniques such as QuaRot Ashkboos et al. (2024) and SpinQuant Liu et al. (2024) use orthogonal transformations to redistribute outliers across channels, while DuQuant Lin et al. (2024a) uses a dual transformation. These methods involve per-token dynamic computation during inference. PrefixQuant Chen et al. (2024a), a recent method, isolates token-wise outliers for efficient per-tensor static quantization, achieving performance that matches or surpasses dynamic methods. Despite quantization addressing computational and storage issues, significant energy consumption during LLM operation remains a barrier to edge deployment due to the power demands of dense matrix multiplication, even with low-bit quantized versions.

### 2.2 ANN-TO-SNN CONVERSION

Existing methods for converting ANNs to SNNs are feasible because the ReLU activation value can be approximated by the firing rate of an IF neuron over a given time window. For ReLU-based ANNs, one-stage conversion methods can be applied without additional optimization Bu et al. (2022). However, these one-stage methods often require a large number of time steps to achieve competitive performance, which significantly diminishes the energy efficiency benefits of SNNs. To address this issue, two-stage methods have been developed, involving the training of a tailored ANN to enable a more effective conversion. These methods further optimize the converted SNN by utilizing techniques such as residual membrane potential Hao et al. (2023a), initial membrane potential shifting Hao et al. (2023b), and local tandem learning rules Yang et al. (2022). While these approaches achieve higher performance with fewer time steps, they necessitate the training of tailored ANNs, which remain computationally intensive, particularly for LLMs.

## 3 OUR NEW ANN-TO-SNN CONVERSION FOR LLM

In this section, we first briefly introduce the conventional ANN-SNN conversion, then give our dual ANN-SNN conversion framework.

## 3.1 CONVENTIONAL ANN-TO-SNN CONVERSION

Conventional ANN-to-SNN conversion methods consider using the Integrate-and-Fire (IF) neuron, whose dynamics can be formulated as

$$\boldsymbol{v}_{IF}^k(t) = \boldsymbol{v}_{IF}^k(t-1) + W^k \theta^{k-1} \boldsymbol{s}_{IF}^{k-1}(t) - \theta^k \boldsymbol{s}_{IF}^k(t), \tag{1}$$

$$\boldsymbol{s}_{IF}^k(t) = H(\boldsymbol{v}_{IF}^k(t) - \theta^k). \tag{2}$$

Here, $\theta^k$ is the threshold, $H(\cdot)$ is the Heaviside function, while the superscript $k$ denotes the layer, $W^k$ denotes the weights between layer $k$ and $k-1$.

The idea of ANN-to-SNN conversion is to relate the ReLU activation of analog neurons in ANNs to the firing rate of spiking neurons in SNNs. By summing the equations over $t = 1, \ldots, T$ and rearranging the terms, we obtain

$$\theta^k \frac{\sum_{t=1}^T \boldsymbol{s}_{IF}^k(t)}{T} = W^k \theta^{k-1} \frac{\sum_{t=1}^T \boldsymbol{s}_{IF}^{k-1}(t)}{T} - \frac{\boldsymbol{v}_{IF}^k(T) - \boldsymbol{v}_{IF}^k(0)}{T}. \tag{3}$$

Let $\phi^k(T) = \theta^k \frac{\sum_{t=1}^T \boldsymbol{s}_{IF}^k(t)}{T}$ to denote the average postsynaptic potential, the above equation can be rewritten as follows

$$\phi^k(T) = W^k \phi^{k-1}(T) - \frac{\boldsymbol{v}_{IF}^k(T) - \boldsymbol{v}_{IF}^k(0)}{T}. \tag{4}$$

On the ANN side, the transformation between layers is given by

$$a^k = \mathcal{A}^k(W^k a^{k-1}), \tag{5}$$

where $\mathcal{A}^k$ is the activation function, which is often set as the ReLU function. Once the activation value $a^k$ is aligned with $\phi^k(T)$ (i.e., $a^k \approx \phi^k(T)$), the ANN can be transformed into an SNN by replacing the ReLU functions with IF neurons and mapping the pre-trained weights and biases onto the same network architecture.

Recently, quantized ANN models have become a popular intermediate representation in ANN-to-SNN conversion. In particular, for ReLU-based ANNs, the ReLU activation can be replaced with a specialized quantization function known as the Quantization-Clip-Floor-Shift (QCFS) function Bu et al. (2022):

$$\mathbf{X}_q^k = \lambda^k \cdot \text{clip}(\lfloor \frac{\mathbf{X}^k}{\lambda^k} + \varphi \rfloor, 0, N), \tag{6}$$

with given quantization steps $N$, scale value $\lambda^k$ and $\varphi = 0.5$. Then, after training (or fine-tuning) the ANN with QCFS as the activation function, and setting $\theta^k = \lambda^k$, $T = N$, and $\boldsymbol{v}^k(0) = \varphi$, the quantized ANN can be easily converted to an SNN model Bu et al. (2022). This pipeline is shown in Figure 1(a).

## 3.2 DUAL ANN-TO-SNN CONVERSION

A common strategy for developing a spiking LLM is to first train a tailored ANN using a conversion-friendly activation function, such as ReLU or QCFS, and then replace the activation function with IF neurons. However, this approach has two major limitations: (1) training a customized LLM specifically for conversion is generally impractical due to the enormous computational cost; and (2) achieving high performance with the conventional ANN-to-SNN conversion typically requires high latency (e.g., > 50 timesteps), which leads to increased energy consumption.

To overcome these challenges, we introduce a novel dual ANN-to-SNN conversion framework that is both training-free and low-latency. Specifically, in the first stage, a widely adopted training-free quantization technique is employed to obtain quantized LLMs. Subsequently, we design a new spiking neuron model that accurately emulates the quantization function, thereby enabling efficient and seamless conversion of quantized LLMs into their spiking counterparts. We also provide a high-leve comparison of two conversion methods in Table 1.

Table 1: Comparison of two ANN-to-SNN conversion methods.

| Method | Neuron | Training Trailored ANN | Latency |
|--------|--------|------------------------|---------|
| Conventional | IF | Yes | High |
| Ours | IS | No | Low |

### 3.2.1 QUANTIZATION FOR LARGE LANGUAGE MODEL

The quantization technique plays a crucial role in deploying large models on resource-constrained devices, such as mobile phones and edge devices, as it significantly reduces model size, enhances inference speed, and decreases power consumption Shao et al. (2024); Lin et al. (2024a). The quantization processes are defined as follows:

$$\mathbf{X}_q^k = \lambda^k \cdot \text{clip} \left( \left\lfloor \frac{\mathbf{X}^k}{\lambda} \right\rceil, -2^{n-1}, 2^{n-1} - 1 \right), \tag{7}$$

where $\mathbf{X}^k$ and $\mathbf{X}_q^k$ denote input and output of the quantization processes, respectively. $\lambda^k$ denotes the scale value, where $n$ represents the quantization bits. In our approach, we employ the static quantization method PrefixQuant Chen et al. (2024a), which eliminates the need to compute scaling factors for each input batch dynamically.

### 3.2.2 APPROXIMATION OF QUANTIZATION FUNCTION WITH INTEGER SPIKING NEURON

We introduce a modified Integer Spiking (IS) neuron—also referred to as the Multi-Hierarchical Threshold (M-HT) neuron Sun et al. (2022); Wang & Zhang (2023); Li & Zeng (2022); Hao et al. (2024)—to emulate the behavior of the quantization function used in PrefixQuant. The dynamics of the proposed IS neuron are defined as follows:

$$\boldsymbol{m}^k(t) = \boldsymbol{v}^k(t-1) + \mathbf{I}^k(t) + \boldsymbol{\alpha}^k(t)\theta^k \ , \boldsymbol{v}^k(t) = \boldsymbol{m}^k(t) - \boldsymbol{s}^k(t)\theta^k, \tag{8}$$

$$\boldsymbol{s}^k(t) = \begin{cases} L, & \boldsymbol{m}(t) \geq L\theta^k \\ l, & l\theta^k \leq \boldsymbol{m}^k(t) < (l+1)\theta^k, \ l = 1, ..., L-1 \\ 0, & \text{otherwise} \end{cases} \tag{9}$$

$$\hat{\boldsymbol{s}}^k(t) = \boldsymbol{s}^k(t) - \boldsymbol{\alpha}^k(t), \tag{10}$$

where $L$ denotes the number of discrete threshold levels, $\mathbf{I}^k(t)$ denotes the input current, $\boldsymbol{\alpha}^k(t) \in [0, L]$ is set by users, which enables the representation of both positive and negative activations in the real output $\hat{\boldsymbol{s}}^k(t)$, thereby extending the expressiveness of SNNs.

For the proposed IS neuron, the total spike output is formally characterized in the following theorem.

**Theorem 1.** *For an IS neuron with $L$-level threshold, after $T > 0$ time-steps, $\forall t \in \{1, \ldots, T\}$, let $\boldsymbol{v}^k(0) = \frac{\theta^k}{2}$, if $\mathbf{I}^k(t)$ falls into one of the three mutually exclusive and exhaustive intervals: $\left(-\infty, -\boldsymbol{\alpha}^k(t)\theta^k\right)$, $\left[-\boldsymbol{\alpha}^k(t)\theta^k, L\theta^k - \boldsymbol{\alpha}^k(t)\theta^k\right)$ or $\left[L\theta^k - \boldsymbol{\alpha}^k(t)\theta^k, +\infty\right)$, we will have:*

$$\sum_{t=1}^{T} \hat{\boldsymbol{s}}^k(t) = clip \left( \left[ \frac{\sum_{t=1}^{T} \mathbf{I}^k(t)}{\theta^k} + \sum_{t=1}^{T} \boldsymbol{\alpha}^k(t) \right], 0, LT \right) - \sum_{t=1}^{T} \boldsymbol{\alpha}^k(t)$$

Then, based on the above result, we show under what conditions our IS neuron can be equivalent to the quantization function.

**Theorem 2.** *Given the quantization bit $n$, for an IS neuron with $L$-level thresholds and time step $T$, let $\boldsymbol{v}^k(0) = \frac{\theta^k}{2}$, $\theta^k = \lambda^k$, $LT = 2^n - 1$, $\boldsymbol{\alpha}^k(t) = \frac{2^{n-1}}{T}$, and $\sum_{t=1}^{T} \mathbf{I}^k(t) = \mathbf{X}^k$, if the following conditions are satisfied: $\forall t \in \{1, \ldots, T\}$, $\mathbf{I}^k(t)$ falls into one of the three mutually exclusive and exhaustive intervals, $\left(-\infty, -\boldsymbol{\alpha}^k(t)\theta^k\right)$, $\left[-\boldsymbol{\alpha}^k(t)\theta^k, L\theta^k - \boldsymbol{\alpha}^k(t)\theta^k\right)$ or $\left[L\theta^k - \boldsymbol{\alpha}^k(t)\theta^k, +\infty\right)$, the outputs of IS neuron can mimic the symmetric quantization function, i.e., $\sum_{t=1}^{T} \hat{\boldsymbol{s}}^k(t)\theta^k = \mathbf{X}_q^k$.*

**Remark 1.** *The above theorem indicates that with a proper setting of $\boldsymbol{v}^k(0)$, $\theta^k$, $L$, $T$ and $\boldsymbol{\alpha}^k(t)$, we can replace the quantization function with our proposed IS neuron. In practical implementations, $L$ (number of threshold levels) and $T$ (time-steps) are usually constrained to be integers. Therefore, equality $LT = 2^n - 1$ rarely holds for arbitrary integer choices of $L$ and $T$ if $T \neq 1$. This implies that*

*the exact equivalence between the IS neuron outputs and the quantization function may not be perfectly achieved in practice. In addition, since $T$ is usually set by $T = 2^j$, where $j \in \{0, 1, \cdots, n-1\}$, we can set $\boldsymbol{\alpha}^k(t) = 2^{n-j-1}$ and $L = \lceil \frac{2^n - 1}{T} \rceil$, therefore, we have $\sum_{t=1}^{T} \hat{\boldsymbol{s}}^k(t)\theta^k \approx \mathbf{X}_q^k$.*

### 3.2.3 ARCHITECTURE OF OUR SPIKING LLM

As a concrete example, we illustrate the conversion process using LLaMA-2 Touvron et al. (2023), as shown in Figure 2, along with its corresponding architecture. It is important to note that replacing the quantization function with IS neurons is only applicable to linear operations. However, the LLaMA model incorporates several nonlinear operations (e.g., activation–activation multiplication, LayerNorm, SiLU activation, and Softmax). To preserve the functional equivalence between the quantized LLMs and the converted spiking LLMs, we adopt the spiking-compatible operations proposed in You et al. (2024). Additional implementation details are provided in the Appendix.

### 3.3 CONVERSION ERROR

In this section, we analyse the conversion error of our method between the source LLM and the converted SNN in each layer in detail.

Similar to conventional ANN-to-SNN conversion, as shown in Bu et al. (2022); Hao et al. (2023a), the dual ANN-SNN conversion also involves three main types of errors: clipping error, quantization error, and unevenness error, all of which contribute to the performance gap between ANNs and SNNs.

**Clipping error.** For an IS neuron with $L$-level thresholds, after $T$ time-steps, the output of IS: $\sum_{t=1}^{T} \hat{\boldsymbol{s}}^k(t)\theta$ is in the range of $\left[ -\sum_{t=1}^{T} \boldsymbol{\alpha}^k(t)\theta^k, LT\theta^k - \sum_{t=1}^{T} \boldsymbol{\alpha}^k(t)\theta^k \right]$. We define the output activation of the ANNs as $a$, and $a_{max}$ as the maximum value in $a$. $a$ can be mapped to: $\text{clip}\left( \theta^k \cdot \left\lfloor \frac{a}{\beta} \right\rfloor, 0, LT\theta^k \right) - \sum_{t=1}^{T} \boldsymbol{\alpha}^k(t)\theta^k$, where $\beta$ represents the actual maximum value of output $a$. Then the output $a \in [\beta, a_{max}]$ of ANNs will be mapped to the same value $(LT\theta^k - \sum_{t=1}^{T} \boldsymbol{\alpha}^k(t)\theta^k)$, the output $a \in (-\infty, 0)$ of ANNs will be mapped to the same value $(-\sum_{t=1}^{T} \boldsymbol{\alpha}^k(t)\theta^k)$, which will cause conversion error called clipping error.

**Quantization error.** The output spike $\hat{s}^k(t)$ is discrete, so the final output $\sum_{t=1}^{T} \hat{\boldsymbol{s}}^k(t)\theta$ is also discrete, while the output activation value $a$ of the ANNs is continuous. Therefore, when mapping $a$ to $\sum_{t=1}^{T} \hat{\boldsymbol{s}}^k(t)\theta^k$, there will be unavoidable quantization error. For example, when the output of ANNs satisfies $a \in \left[ l\lambda^k, (l+1)\lambda^k \right), l = 0, 1, ..., LT-1$, the corresponding mapped value of $\sum_{t=1}^{T} \hat{\boldsymbol{s}}^k(t)$ will be $\left( l - \sum_{t=1}^{T} \boldsymbol{\alpha}^k(t) \right) \theta^k$.

**Unevenness error.** Although we set $LT = 2^n - 1$, $\sum_{t=1}^{T} \mathbf{I}^k(t) = \mathbf{X}^k$, $\theta^k = \lambda^k$ and $\boldsymbol{v}^k(0) = \frac{\theta^k}{2}$, changes in spike timing can lead to the generation of more or fewer spikes than expected. This error, caused by the unevenness of the input currents, is referred to as unevenness error. The accumulation of such errors across multiple neurons in the SNN model can result in discrepancies in the output compared to the QANN model's output. This discrepancy is defined as the unevenness error of the SNN model. Here, we define the unevenness error of the SNN models as follows:

**Definition 1** (Unevenness error of SNN model). *Given a QANN model $g$ with input $x$, its corresponding output is denoted by $\bar{y} = g(x)$. Consider the converted SNN model $f$ obtained via our dual ANN-SNN conversion method. If the input to the SNN is a sequence $(\hat{x}(1), \hat{x}(2), \ldots, \hat{x}(T))$ satisfying $\sum_{t=1}^{T} \hat{x}(t) = x$, then the corresponding output of the SNN is given by $(\hat{y}(1), \hat{y}(2), \ldots, \hat{y}(T)) = f(\hat{x}(1), \hat{x}(2), \ldots, \hat{x}(T))$. The* unevenness error *between the SNN and the QANN models can then be explicitly defined as $\left\| \sum_{t=1}^{T} \hat{y}(t) - \bar{y} \right\|$.*

In order to more intuitively demonstrate these errors in ANN-to-SNN conversion, we converted the Llama-2-7B model to an SNN through our conversion method. Figure 3 presents the conversion error at each layer of every stage by computing the MSE loss. It shows that unevenness error plays mainly character in ANN-to-SNN conversion.

### 3.3.1 ANALYSIS OF LAYER-WISE ERRORS

We first give Assumption 1, then build the upper bound to the conversion error with our conversion method in Theorem 3.

**Assumption 1** (Bounded propagation of layer–wise errors). *Let $k = 1, 2, ..., K$ be the layer index in the QANN model, and $g^k$ be the function of k-th layer for QANN. We assume that there exists a corresponding constant $\rho^k$ for $g^k$, such that, for any input $\bar{x}_1^k$ and $\bar{x}_2^k$, we have*

$$\left\| g^k \left( \bar{x}_1^k \right) - g^k \left( \bar{x}_2^k \right) \right\| \leq \rho^k \left\| \bar{x}_1^k - \bar{x}_2^k \right\|. \quad (11)$$

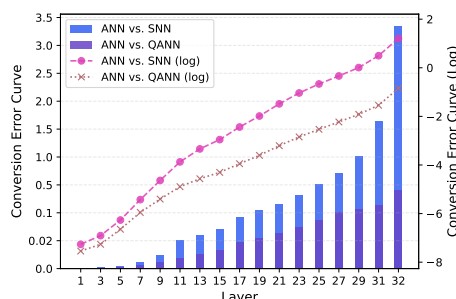

Figure 3: The layer-wise MSE loss between ANN vs. QANN and ANN vs. SNN ($T = 2$) on LLaMA-2-7B. The former can represent clipping and quantization error, and the difference between the two can measure the magnitude of the unevenness error.

**Remark 2.** *For QANN model, the bounded propagation of layer-wise errors satisfies a Lipschitz-like condition consistent with the classical Lipschitz continuity assumption. This constraint reinforces the theoretical stability of QANNs and aligns with assumptions commonly adopted in prior research on neural network quantization Li et al. (2017); Hou et al. (2019); Chen et al. (2021); Zhang et al. (2022).*

**Theorem 3.** *Let $k = 1, 2, ..., K$ be the layer index and $\rho^k$ is the corresponding layer-wise lipschitz constant, given an ANN model $h$ with input $x$, its corresponding output is denoted by $y = h(x)$, and a QANN model $g$ with input $x$, its corresponding output is denoted by $\bar{y} = g(x)$. Consider the converted SNN model $f$ obtained via our dual ANN-SNN conversion method. If the input to the SNN is a sequence $(\hat{x}(1), \hat{x}(2), \ldots, \hat{x}(T))$ satisfying $\sum_{t=1}^{T} \hat{x}(t) = x$, then the corresponding output of the SNN is given by $(\hat{y}(1), \hat{y}(2), \ldots, \hat{y}(T)) = f(\hat{x}(1), \hat{x}(2), \ldots, \hat{x}(T))$. The conversion error can be bounded as follows:*

$$\left\| \sum_{t=1}^{T} \hat{y}(t) - y \right\| \leq \sum_{k=1}^{K} \left( \prod_{\tau=k+1}^{K} \rho^\tau \right) \left\| \sum_{t=1}^{T} \hat{y}^k(t) - g^k(\sum_{t=1}^{T} \hat{x}^k(t)) \right\| + \sum_{k=1}^{K} \left( \prod_{\tau=k+1}^{K} \rho^\tau \right) \left\| g^k(x^k) - y^k \right\|.$$

**Remark 3.** *The conversion error in the SNN model comprises clipping and quantization error, as well as unevenness errors, which accumulate progressively across layers, an observation corroborated by experimental results (Figure 3). The theorem implies that Layer-wise calibration of these errors can effectively mitigate the overall conversion error.*

### 3.4 LAYER-WISE CALIBRATION

Due to clipping and quantization error caused by quantization and unevenness error caused by our dual ANN-to-SNN conversion, the performance of the SNN model we obtained is still not ideal if $T > 1$. To address this problem, we propose a layer-wise calibration method to minimize the upper bound in Theorem 3, thereby reducing the gap between the source ANN and the converted SNN. We have reason to believe that the weights obtained from QANN are good enough, leading us to freeze the weights and optimize a finite set of neuronal thresholds and initial membrane potentials.

Our calibration target can be formulated by the following formula: $\min_{\theta^k, v^k(0)} \left\| \sum_t^T \hat{y}^k(t) - y^k \right\|$, where $\theta^k$ and $v^k(0)$ are the threshold and initial membrane potential of IS neuron, respectively. The subsequent experiments confirm that the performance of the post-calibration SNN is on par with that of the QANN.

## 4 EXPERIMENTS

### 4.1 EXPERIMENTAL SETUPS

**Baselines.** We compare the performance of our method[1] with PrefixQuant Chen et al. (2024a) and DuQuant Lin et al. (2024a). We also compare our method with the ANN-to-SNN conversion and the

---

[1]All source code required for conducting experiments will be made publicly available upon publication of the paper.

calibration involving all parameters. Experiments are conducted under the W6A6 (6-bit weights and 6-bit activations) quantization setting.

**Models and Datasets.** We evaluate the performance of our method on the LLaMA-2 and LLaMA-3 families. Following the previous literature Son et al. (2024); Shao et al. (2024); Lin et al. (2024b), we evaluate all the methods on five zero-shot reasoning tasks, including PIQA Bisk et al. (2020), ARC Clark et al. (2018), HellaSwag Zellers et al. (2019), and WinoGrande Sakaguchi et al. (2021). All accuracies are measured using lm-evaluation-harness: v0.4.2[2] Sutawika et al. (2024). We report the accuracy (acc) for WinoGrande and accuracy norm (acc_norm) for HellaSwag, Arc Challenge (ArcC), Arc Easy (ArcE), and PIQA, following Qserve Lin et al. (2024b). For perplexity (PPL), we provide the results for WikiText2 Merity et al. (2016). All experiments are conducted on servers equipped with multiple 80GB NVIDIA A100 GPUs.

## 4.2 EVALUATE PERFORMANCE ON DIFFERENT MODELS

In this subsection, we conduct a detailed evaluation of the performance of our proposed method across two different models: LLaMA-2-7B and LLaMA-3-8B. We evaluate the performance of our method using different $T$, such as $T = 2$, $T = 4$, and $T = 8$. The goal is to compare the performance of our method with existing approaches, highlighting the advantages of our method in terms of effectiveness. We evaluate various performance metrics, such as accuracy and perplexity, to provide a comprehensive comparison. Table 2 demonstrates that our proposed conversion framework yields high-performing SNN models, with the layer-wise calibration method substantially contributing to performance improvements. Meanwhile, the significant performance gap between the uncalibrated SNNs and their quantized counterparts, as observed in Table 2, further corroborates our observation that the unevenness error constitutes the dominant source of degradation. Moreover, as time-step $T$ increases, the performance degrades correspondingly. We attribute this phenomenon to the growing unevenness error introduced by the larger time-step.

Table 2: Results on LLaMA-2-7B and LLaMA-3-8B. We report acc for WinoGrande and acc_norm for HellaSwag, ArcC, ArcE, and PIQA. For PPL, we report the results on WikiText2. "Conversion" denotes the SNN performance before calibration. "Avg. Acc." indicates the average zero-shot accuracy on five reasoning tasks.

| Model | Method | Presicion | WinoGrande | HellaSwag | ArcC | ArcE | PIQA | Avg. Acc. | PPL (↓) |
|---|---|---|---|---|---|---|---|---|---|
|  | Baseline | FP16 | 69.22 | 76.00 | 46.25 | 74.62 | 79.11 | 69.04 | 5.47 |
|  | PrefixQuant | W6A6 | 70.17 | 75.70 | 45.99 | 74.41 | 77.26 | 68.70 | 5.76 |
|  | DuQuant | W6A6 | 67.88 | 72.64 | 40.53 | 53.07 | 77.15 | 62.25 | 5.53 |
|  | Conversion | W6A6, T=1 | 70.17 | 75.70 | 45.99 | 74.41 | 77.26 | 68.70 | 5.76 |
|  | Ours | W6A6, T=1 | 69.61 | 76.22 | 46.08 | 73.82 | 78.24 | 68.79 | 5.61 |
| 2-7B | Conversion | W6A6, T=2 | 64.72 | 59.02 | 37.37 | 65.19 | 73.67 | 59.99 | 12.42 |
|  | Ours | W6A6, T=2 | 68.67 | 74.91 | 44.45 | 72.39 | 77.86 | 67.65 | 7.39 |
|  | Conversion | W6A6, T=4 | 59.19 | 41.14 | 28.67 | 53.49 | 68.82 | 50.26 | 97.76 |
|  | Ours | W6A6, T=4 | 68.35 | 74.32 | 43.34 | 72.18 | 77.04 | 67.04 | 9.71 |
|  | Conversion | W6A6, T=8 | 53.51 | 31.61 | 25.43 | 33.00 | 55.55 | 39.82 | 319.36 |
|  | Ours | W6A6, T=8 | 66.54 | 73.96 | 42.32 | 70.37 | 76.99 | 66.03 | 12.03 |
|  | Baseline | FP16 | 72.22 | 79.37 | 49.06 | 77.48 | 80.52 | 71.73 | 6.14 |
|  | PrefixQuant | W6A6 | 71.11 | 78.04 | 48.32 | 75.13 | 77.64 | 70.24 | 6.90 |
|  | DuQuant | W6A6 | 67.88 | 72.64 | 40.53 | 53.07 | 77.15 | 62.25 | 6.27 |
|  | Conversion | W6A6, T=1 | 71.11 | 78.04 | 48.32 | 75.13 | 77.64 | 70.24 | 6.90 |
|  | Ours | W6A6, T=1 | 73.01 | 77.67 | 51.88 | 76.47 | 79.33 | 71.67 | 6.66 |
| 3-8B | Conversion | W6A6, T=2 | 58.96 | 41.87 | 26.79 | 49.92 | 66.65 | 48.83 | 29.97 |
|  | Ours | W6A6, T=2 | 70.64 | 75.99 | 50.85 | 70.75 | 76.93 | 69.03 | 9.07 |
|  | Conversion | W6A6, T=4 | 50.20 | 32.81 | 22.78 | 39.44 | 60.23 | 41.09 | 81.38 |
|  | Ours | W6A6, T=4 | 66.22 | 72.03 | 48.81 | 71.59 | 77.42 | 67.21 | 11.67 |
|  | Conversion | W6A6, T=8 | 50.51 | 29.91 | 19.8 | 32.79 | 56.58 | 37.91 | 190.63 |
|  | Ours | W6A6, T=8 | 62.98 | 63.96 | 46.33 | 69.28 | 76.28 | 63.76 | 18.93 |

---

[2] https://doi.org/10.5281/zenodo.10829972

## 4.3 Evaluate Performance of Our Framework with Different Learnable Parameter Sizes

In this subsection, we investigate the performance of our proposed method under varying sizes of learnable parameters. The size of the learnable parameters is a crucial factor that directly impacts the performance of the model. By exploring different parameter sizes, we aim to understand how our framework adapts to different conversion settings. Specifically, the size of the learnable parameters, including the thresholds and initial membrane potentials, depends on the number of activation groups. Given the constant activation dimensionality, there exists an inverse relationship between the group size and the learnable parameter size. We evaluate the performance of our framework based on LLaMA-2-7B under $T = 2$ using different group sizes and record the corresponding learnable parameter sizes in Table 3. The performance does not vary significantly across different parameter sizes, which demonstrates the strong adaptability of our framework to diverse conversion settings.

Table 3: Results of our framework under different learnable parameter sizes on LLaMA-2-7B. "Avg. Acc." indicates the average zero-shot accuracy on five reasoning tasks. For PPL, we report the results on WikiText2. We vary the activation group size to configure different learnable parameter sizes. "-1" indicates that the group size is set to the activation dimensionality. "Param Size" denotes the learnable parameter size at each layer.

| Group Size | Param Size | Avg. Acc. | PPL ($\downarrow$) |
|---|---|---|---|
| 256 | 0.194K | 67.40 | 7.11 |
| 64 | 0.467K | 67.31 | 9.17 |
| 16 | 1.559K | 67.03 | 6.89 |
| 4 | 5.927K | 66.70 | 7.15 |
| 1 | 23.399K | 65.46 | 7.20 |
| -1 (Ours) | 0.107K | 67.65 | 7.39 |

## 4.4 Comparison with Layer-wise Weight Calibration

In this subsection, we compare our layer-wise calibration method, which only adjusts thresholds and initial membrane potentials, with layer-wise weight calibration in terms of performance under $T = 2$. The results in Table 4 demonstrate the parameter efficiency of our method. Our calibration method achieves comparable performance while using significantly fewer learnable parameters than weight calibration. These results highlight the effectiveness and parameter efficiency of our method, underscoring its advantage over traditional weight calibration in SNN conversion.

## 5 Conclusion

We propose a dual ANN-to-SNN conversion framework that offers a promising solution to the challenges of deploying LLMs on edge devices. By leveraging quantized LLMs and eliminating the need for a specially trained ANN, our approach significantly reduces computational costs and complexity. Through the introduction of a parameter-efficient layer-wise calibration technique, our method effectively addresses the issue of unevenness error, ensuring that the performance of the converted SNN closely matches that of the original LLM. Our theoretical analysis and extensive experiments on LLaMA models confirm the efficacy of our framework, showcasing substantial improvements in accuracy and making it a viable option for the edge-based deployment of large-scale models.

Table 4: Comparison of our calibration method and weight fine-tuning. "Avg. Acc." indicates the average zero-shot accuracy on five reasoning tasks. For PPL, we report the results on WikiText2. "Param Size" denotes the learnable parameter size at each layer.

| Model | Type | Param Size | Avg. Acc. | PPL ($\downarrow$) |
|---|---|---|---|---|
| 2-7B | Weight | 202.375M | 66.39 | 6.37 |
| | Ours | 0.107K | 67.65 | 7.39 |
| 3-8B | Weight | 218.103M | 68.65 | 8.04 |
| | Ours | 0.107K | 69.03 | 9.07 |

## Ethics Statement

All participants in this work, as well as the paper submission, adhere to the ICLR Code of Ethics ( https://iclr.cc/public/CodeOfEthics).

## REPRODUCIBILITY STATEMENT

We affirm that the results of this work are fully reproducible. The source code will be publicly released after publication of the paper.

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

## A    USE OF LLMS

In this work, LLMs are employed solely for polishing or grammar checking text that is originally written by us.

## B    NOTIONS

We provide the notation used in the paper in Table 5.

## C    ADDITIONAL EXPERIMENTS

### C.1    EVALUATE CONVERSION ERROR WITH DIFFERENT $T$

In this subsection, we evaluate the conversion error before and after calibration by computing the MSE loss. The conversion error serves as a reliable indicator for quantifying the gap between an ANN and its converted SNN. By statistically analyzing and comparing the layer-wise errors before and after calibration, the effect of our proposed calibration method can be clearly demonstrated. As shown in Figure 4, the error exhibits a layer-wise accumulation effect, which becomes increasingly pronounced in the deeper layers. Moreover, our proposed calibration method effectively mitigates this error accumulation. This is evidenced by the post-calibration error curve lying significantly below that of the uncalibrated SNN, particularly when $T = 8$, thereby enabling a more reliable ANN-to-SNN conversion.

Table 5: Symbol Definitions

| Symbol | Definition |
|---|---|
| $k$ | Layer index |
| $K$ | The number of layer |
| $\lfloor \cdot \rfloor$ | Floor operation |
| $\lfloor \cdot \rceil$ | Round operation |
| $\mathbf{Q}_q$ | Quantized query $\mathbf{Q}$ |
| $\mathbf{K}_q$ | Quantized key $\mathbf{K}$ |
| $\mathbf{V}_q$ | Quantized value $\mathbf{V}$ |
| $\mathbf{Q}_{s,t}$ | Spiking query $\mathbf{Q}$ |
| $\mathbf{K}_{s,t}$ | Spiking key $\boldsymbol{s}_{IF}^k(t)$ |
| $\mathbf{V}_{s,t}$ | Spiking value $\mathbf{V}$ |
| $\mathbf{X}_q$ | Dequantized tensor |
| $\mathbf{X}_{\text{INT}}$ | Quantized tensor |
| $\mathbf{X}$ | Full-precision tensor |
| $\lambda$ | Scale in quantization |
| $n$ | Quantization bits in quantization |
| $\gamma,\ \beta$ | Clipping factors |
| $\boldsymbol{m}(t)$ | Membrane potential before firing in IS |
| $\boldsymbol{v}(t)$ | Membrane potential after firing in IS |
| $\boldsymbol{s}(t)$ | Output spikes in IS |
| $L$ | Number of discrete threshold levels |
| $\theta$ | Threshold |
| $t$ | Time step |
| $T$ | Total time step |
| $\mathbf{I}(t)$ | Input current |
| $\phi(\cdot)$ | The Softmax, Layernorm and SiLU activation |

## D THE EQUIVALENCE BETWEEN IS NEURONS AND QUANTIZATION FUNCTION

*Proof.* Proof of Theorem 1:

**Case 1:** If $\forall t, \mathbf{I}^k(t) < -\alpha^k(t)\theta^k$. No spikes are released at this case: $\sum_{t=1}^{T} \hat{\boldsymbol{s}}^k(t) = -\sum_{t=1}^{T} \alpha^k(t)$

**Case 2:** If $\forall t, \mathbf{I}^k(t) \geq L\theta^k - \alpha^k(t)\theta^k$. This case puts out L spikes per time step: $\sum_{t=1}^{T} \hat{\boldsymbol{s}}^k(t) = LT - \sum_{t=1}^{T} \alpha^k(t)$

**Case 3:** If $\forall t, -\alpha^k(t)\theta^k \leq \mathbf{I}^k(t) < L\theta^k - \alpha^k(t)\theta^k$.

Integrating the input and output membrane potentials gives:

$$\boldsymbol{v}^k(t) = \boldsymbol{v}^k(t-1) + \mathbf{I}^k(t) - \boldsymbol{s}^k(t)\theta^k + \alpha^k(t)\theta^k$$

Organize the above equation and add up with respect to t and then take the average:

$$\frac{\boldsymbol{v}^k(T) - \boldsymbol{v}^k(0)}{T} = \frac{\sum_{t=1}^{T} \mathbf{I}^k(t)}{T} - \frac{\sum_{t=1}^{T} \boldsymbol{s}^k(t)\theta^k}{T} + \alpha^k(t)\theta^k \tag{12}$$

Then:

$$\frac{\sum_{t=1}^{T} \boldsymbol{s}^k(t)\theta^k}{T} = \frac{\sum_{t=1}^{T} \mathbf{I}^k(t)}{T} - \frac{\boldsymbol{v}^k(T) - \boldsymbol{v}^k(0)}{T} + \alpha^k(t)\theta^k \tag{13}$$

Then:

$$\sum_{t=1}^{T} \boldsymbol{s}^k(t) = \frac{\sum_{t=1}^{T} \mathbf{I}^k(t)}{\theta^k} - \frac{\boldsymbol{v}^k(T) - \boldsymbol{v}^k(0)}{\theta^k} + \sum_{t=1}^{T} \alpha^k(t) \tag{14}$$

Since $\sum_{t=1}^{T} \boldsymbol{s}^k(t)$ is an integer:

$$\sum_{t=1}^{T} \boldsymbol{s}^k(t) = \left\lfloor \frac{\sum_{t=1}^{T} \mathbf{I}^k(t) + \boldsymbol{v}^k(0)}{\theta^k} + \frac{LT+1}{2} \right\rfloor - \left\lfloor \frac{\boldsymbol{v}^k(T)}{\theta^k} \right\rfloor \tag{15}$$

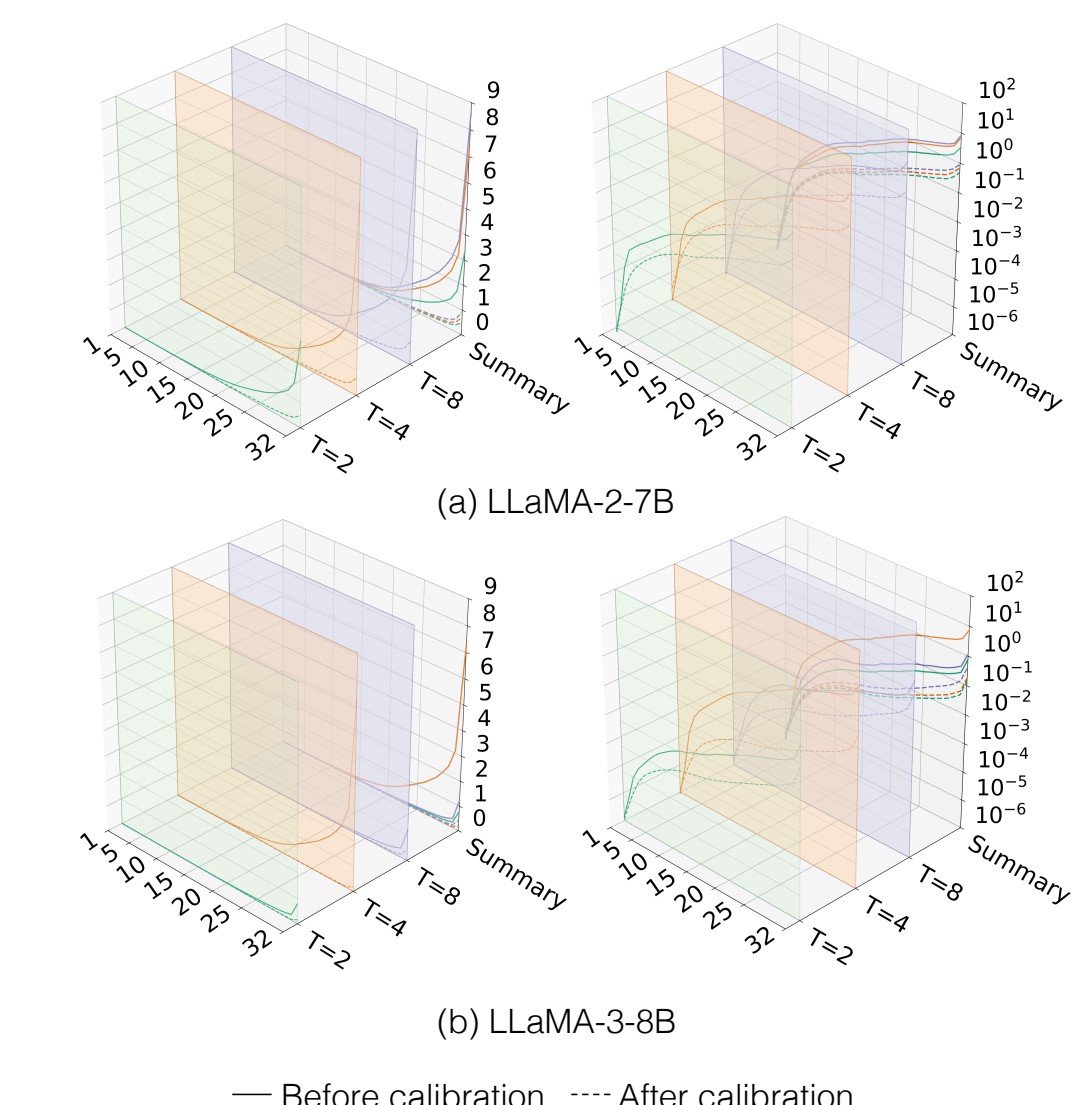

(a) LLaMA-2-7B

(b) LLaMA-3-8B

—— Before calibration    ---- After calibration

Figure 4: The layer-wise error on LLaMA-2-7B and LLaMA-3-8B, which are measured using the MSE loss. The left shows the error curves under different $T$, while the right presents the logarithmic error curves. "Summary" aggregates the (logarithmic) error curves across all $T$.

Due to $-\alpha^k(t)\theta^k \leq \mathbf{I}^k(t) < L\theta^k - \alpha^k(t)\theta^k$ and $\boldsymbol{v}^k(t) = \boldsymbol{m}^k(t) - \boldsymbol{s}^k(t)\theta^k$:

$\forall t \in [0, T)$, if $\boldsymbol{v}^k(0) = \frac{\theta^k}{2}$, $\boldsymbol{v}^k(t) \in [0, \theta^k)$, $m^k(t+1) = \boldsymbol{v}^k(t) + \mathbf{I}^k(t) + \alpha^k(t)\theta^k \in [0, (L+1)\theta^k)$, thus $\boldsymbol{v}^k(t+1) = \boldsymbol{m}^k(t+1) - \boldsymbol{s}^k(t+1)\theta^k \in [0, \theta^k)$, then:

$$\boldsymbol{v}^k(T) \in [0, \theta^k) \tag{16}$$

Then:

$$\sum_{t=1}^{T} \boldsymbol{s}^k(t) = \left\lfloor \frac{\sum_{t=1}^{T} \mathbf{I}^k(t) + \boldsymbol{v}^k(0)}{\theta^k} + \sum_{t=1}^{T} \alpha^k(t) \right\rfloor$$

$$\sum_{t=1}^{T} \boldsymbol{s}^k(t) = \left\lfloor \frac{\sum_{t=1}^{T} \mathbf{I}^k(t) + \boldsymbol{v}^k(0)}{\theta^k} + \sum_{t=1}^{T} \alpha^k(t) \right\rfloor$$

$$\sum_{t=1}^{T} \boldsymbol{s}^k(t) = \text{clip} \left( \left\lfloor \frac{\sum_{t=1}^{T} \mathbf{I}^k(t) + \boldsymbol{v}^k(0)}{\theta^k} + \sum_{t=1}^{T} \alpha^k(t) \right\rfloor, 0, LT \right)$$

$$\sum_{t=1}^{T} \hat{\boldsymbol{s}}^k(t) = \text{clip} \left( \left\lfloor \frac{\sum_{t=1}^{T} \mathbf{I}^k(t) + \boldsymbol{v}^k(0)}{\theta^k} + \sum_{t=1}^{T} \alpha^k(t) \right\rfloor, 0, LT \right)$$

$$- \sum_{t=1}^{T} \alpha^k(t)$$

Let $\frac{\boldsymbol{v}^k(0)}{\theta^k} = 0.5$, then:

$$\sum_{t=1}^{T} \hat{\boldsymbol{s}}^k(t) = \text{clip} \left( \left\lfloor \frac{\sum_{t=1}^{T} \mathbf{I}^k(t)}{\theta^k} + \sum_{t=1}^{T} \alpha^k(t) \right\rceil, 0, LT \right) - \sum_{t=1}^{T} \alpha^k(t)$$

$\square$

*Proof.* Proof of Theorem 2:

When $T = 1$, IS and quantization function are equivalent.

When $T \geq 2$:

Let:

$$\theta^k = \lambda^k$$

$$LT = 2^n - 1$$

$$\alpha^k(t) = \frac{LT + 1}{2T} = \frac{2^{n-1}}{T}$$

$$\sum_{t=1}^{T} \mathbf{I}^k(t) = \mathbf{X}^k$$

Then, in three cases:

**Case 1:** If $\forall t, \mathbf{I}^k(t) < -\alpha^k(t)\theta^k$:

$$\sum_{t=1}^{T} \hat{\boldsymbol{s}}^k(t)\theta^k = -\sum_{t=1}^{T} \alpha^k(t)\theta^k = -2^{n-1} \cdot \lambda^k = \mathbf{X}_q^k \tag{17}$$

**Case 2:** If $\forall t, \mathbf{I}^k(t) \geq L\theta^k - \alpha^k(t)\theta^k$:

$$\sum_{t=1}^{T} \hat{\boldsymbol{s}}^k(t)\theta^k = LT\theta^k - \sum_{t=1}^{T} \alpha^k(t)\theta^k = (2^{n-1} - 1) \cdot \lambda^k = \mathbf{X}_q^k \tag{18}$$

**Case 3:** If $\forall t, -\alpha^k(t)\theta^k \leq \mathbf{I}^k(t) < L\theta^k - \alpha^k(t)\theta^k$:

$$\sum_{t=1}^{T} \hat{\boldsymbol{s}}^k(t)\theta^k$$

$$= \theta^k \left[ \text{clip} \left( \left\lfloor \frac{\sum_{t=1}^{T} \mathbf{I}^k(t)}{\theta^k} + \sum_{t=1}^{T} \alpha^k(t) \right\rceil, 0, LT \right) - \sum_{t=1}^{T} \alpha^k(t) \right]$$

$$= \lambda^k \left[ \text{clip} \left( \left\lfloor \frac{\mathbf{X}^k}{\lambda^k} + 2^{n-1} \right\rceil, 0, 2^n - 1 \right) - 2^{n-1} \right]$$

$$= \lambda^k \cdot \text{clip} \left( \left\lfloor \frac{\mathbf{X}^k}{\lambda^k} \right\rceil, -2^{n-1}, 2^{n-1} - 1 \right) = \mathbf{X}_q^k$$

$\square$

# E  ANALYSIS OF LAYER-WISE ERRORS

*Proof.* The proof of Theorem 3:
When we use the ANN as a base model to calibrate the SNN model, the output error between ANN and SNN models:

$$\left\| \sum_{t=1}^{T} \hat{y}^K(t) - y^K \right\|$$

$$= \left\| \sum_{t=1}^{T} \hat{y}^K(t) - \bar{y}^K + \bar{y}^K - y^K \right\|$$

$$\leq \underbrace{\left\| \sum_{t=1}^{T} \hat{y}^K(t) - \bar{y}^K \right\|}_{a)} + \underbrace{\left\| \bar{y}^K - y^K \right\|}_{b)}.$$

The first term represents the error between the SNN and the QANN, if each neuron satisfies $LT = 2^n - 1$, $\sum_{t=1}^{T} \mathbf{I}^k(t) = \mathbf{X}^k$, $\theta^k = \lambda^k$ and $\boldsymbol{v}^k(0) = \frac{\theta^k}{2}$, this error is the unevenness error. The second term represents the error caused by the quantized weights, which consists of clip error and quantization error.

Therefore, our calibration mainly targets the error between SNN and QANN models. Next, for the layer $k$, we analyze the layer-wise unevenness error, as well as the layer-wise clip and quantization errors.

For a), we analyze the layer-wise unevenness error in layer $k$:

$$\left\| \sum_{t=1}^{T} \hat{y}^k(t) - \bar{y}^k \right\|$$

$$= \left\| \sum_{t=1}^{T} f^k(\hat{x}^k(1), \hat{x}^k(2), ..., \hat{x}^k(T)) - g^k(\bar{x}^k) \right\|$$

$$\leq \left\| \sum_{t=1}^{T} f^k(\hat{x}^k(1), \hat{x}^k(2), ..., \hat{x}^k(T)) - g^k(\sum_{t=1}^{T} \hat{x}^k(t)) \right\|$$

$$+ \left\| g^k(\sum_{t=1}^{T} \hat{x}^k(t)) - g^k(\bar{x}^k) \right\|$$

$$= \left\| \sum_{t=1}^{T} \hat{y}^k(t) - g^k(\sum_{t=1}^{T} \hat{x}^k(t)) \right\| + \left\| g^k(\sum_{t=1}^{T} \hat{x}^k(t)) - g^k(\bar{x}^k) \right\|.$$

The first term corrects the input of the layer $k$ of the QANN to match the input of the SNN, thereby focusing on the analysis of the unevenness error in the layer $k$. The second term represents the expected error obtained when the accumulated total error from the layer 1 to the layer $k-1$ passes through the layer $k$, reflecting the accumulation effect of the errors from the previous $k-1$ layers.

Under Assumption 1, we continue analyze the layer-wise unevenness error.

$$\left\| \sum_{t=1}^{T} \hat{y}^k(t) - \bar{y}^k \right\|$$

$$\leq \left\| \sum_{t=1}^{T} \hat{y}^k(t) - g^k(\sum_{t=1}^{T} \hat{x}^k(t)) \right\| + \left\| g^k(\sum_{t=1}^{T} \hat{x}^k(t)) - g^k(\bar{x}^k) \right\|$$

$$\leq \left\| \sum_{t=1}^{T} \hat{y}^k(t) - g^k(\sum_{t=1}^{T} \hat{x}^k(t)) \right\| + \rho^k \left\| \sum_{t=1}^{T} \hat{x}^k(t) - \bar{x}^k \right\|$$

$$= \left\| \sum_{t=1}^{T} \hat{y}^k(t) - g^k(\sum_{t=1}^{T} \hat{x}^k(t)) \right\| + \rho^k \left\| \sum_{t=1}^{T} \hat{y}^{k-1}(t) - \bar{y}^{k-1} \right\|$$

$$\leq \sum_{j=1}^{k} \left( \prod_{\tau=j+1}^{k} \rho^\tau \right) \left\| \sum_{t=1}^{T} \hat{y}^j(t) - g^j(\sum_{t=1}^{T} \hat{x}^j(t)) \right\|$$

$$+ \left( \prod_{\tau=1}^{k} \rho^\tau \right) \left\| \sum_{t=1}^{T} \hat{x}^1(t) - \bar{x}^1 \right\|$$

$$= \sum_{j=1}^{k} \left( \prod_{\tau=j+1}^{k} \rho^\tau \right) \left\| \sum_{t=1}^{T} \hat{y}^j(t) - g^j(\sum_{t=1}^{T} \hat{x}^j(t)) \right\|.$$

For b), similar to the analysis of layer-wise unevenness error, we analyze the layer-wise clip and quantization errors in layer $k$:

$$\left\| \bar{y}^k - y^k \right\|$$

$$= \left\| g^k(\bar{x}^k) - h^k(x^k) \right\|$$

$$= \left\| g^k(\bar{x}^k) - g^k(x) + g^k(x^k) - h^k(x^k) \right\|$$

$$\leq \left\| g^k(\bar{x}^k) - g^k(x) \right\| + \left\| g^k(x^k) - h^k(x^k) \right\|$$

$$\leq \sum_{j=1}^{k} \left( \prod_{\tau=j+1}^{k} \rho^\tau \right) \left\| g^j(x^j) - y^j \right\| + \left( \prod_{\tau=1}^{k} \rho^\tau \right) \left\| \bar{x}^1 - x^1 \right\|$$

$$\overset{(1)}{=} \sum_{j=1}^{k} \left( \prod_{\tau=j+1}^{k} \rho^\tau \right) \left\| g^j(x^j) - y^j \right\|, \tag{19}$$

where (1) is because the input of ANN and QANN model are equal the layer 1.
Considering the final output of the layer $K$, the relationship between the total error of the entire SNN model and the layer-wise errors can be expressed as:

$$\left\| \sum_{t=1}^{T} \hat{y}^K(t) - y^K \right\|$$

$$\leq \sum_{k=1}^{K} \left( \prod_{\tau=k+1}^{K} \rho^\tau \right) \left\| \sum_{t=1}^{T} \hat{y}^k(t) - g^k(\sum_{t=1}^{T} \hat{x}^k(t)) \right\|$$

$$+ \sum_{k=1}^{K} \left( \prod_{\tau=k+1}^{K} \rho^\tau \right) \left\| g^k(x^k) - y^k \right\| \tag{20}$$

The left term $\left\| \sum_{t=1}^{T} \hat{y}^K(t) - y^K \right\|$ represents the output error of the entire SNN model compared to the ANN model; the first term in the right $\left\| \sum_{t=1}^{T} \hat{y}^k(t) - g^k(\sum_{t=1}^{T} \hat{x}^k(t)) \right\|$ represents the layer-wise unevenness error in layer $k$; the second term in the right $\left\| g^k(x^k) - y^k \right\|$ represents layer-wise clip and quantization errors in layer $k$. $\qquad\square$

## F  IMPLEMENT IS NEURON ON BINARY CHIP

Here, we first give the relation between IS neuron and IF neuron.

**Lemma 1.** *Assume a continuous $T$-step input current $\mathbf{I}^k(1), \ldots, \mathbf{I}^k(T)$, for a IS neuron with $L$-level threshold, when $\forall t \in [1, T]$, $\mathbf{I}^k(t) \in [-\alpha^k(t)\theta^k, L\theta^k - \alpha^k(t)\theta^k)$ and $\boldsymbol{v}^k(0) \in [0, \theta^k)$, we will have $\boldsymbol{v}^k(T) \in [0, \theta^k)$.*

*Proof.* Proof of Lemma 1:
$\forall t \in [0, T)$, if $\boldsymbol{v}^k(t) \in [0, \theta^k)$, as $\boldsymbol{m}^k(t+1) = \boldsymbol{v}^k(t) + \mathbf{I}^k(t) + \alpha^k(t)\theta^k$, we have $\boldsymbol{m}^k(t+1) \in [0, (L+1)\theta^k)$. Therefore, after the firing process $\boldsymbol{v}^k(t+1) = \boldsymbol{m}^k(t+1) - \boldsymbol{s}^l(t)\theta^k$, one can note that $\boldsymbol{v}^k(t+1) \in [0, \theta^k)$. According to the idea of mathematical induction, if we directly set $\boldsymbol{v}^k(0) \in [0, \theta^k)$, we can have $\boldsymbol{v}^k(T) \in [0, \theta^k)$ □

**Lemma 2.** *For all $t \in [1, T]$, if $\boldsymbol{v}^l(t-1) \in [0, \theta^l]$, the effect of inputting current $\mathbf{I}^k(t)$ into an IS neuron with $L$-level threshold at the $t$-th time-step is equivalent to continuously inputting uniform current $\mathbf{I}^k(t)/L$ for $L$ time-steps into an IF neuron with $\boldsymbol{v}_{IF}^k(0) = \boldsymbol{v}^k(t-1)$, i.e.*

$$\hat{\boldsymbol{s}}^k(t) = \sum_{j=1}^{L} \boldsymbol{s}_{IF}^k(j). \tag{21}$$

*Proof.* Proof of Lemma 2:
**Case 1:** If $\forall t, \mathbf{I}^k(t) < -\alpha^k(t)\theta^k$:

$$\hat{\boldsymbol{s}}^k(t) = \sum_{j=1}^{L} \boldsymbol{s}_{IF}^k(j) = 0. \tag{22}$$

**Case 2:** If $\forall t, \mathbf{I}^k(t) \geq L\theta^k - \alpha^k(t)\theta^k$:

$$\hat{\boldsymbol{s}}^k(t) = \sum_{j=1}^{L} \boldsymbol{s}_{IF}^k(j) = L. \tag{23}$$

**Case 3:** If $\forall t, -\alpha^k(t)\theta^k \leq \mathbf{I}^k(t) < L\theta^k - \alpha^k(t)\theta^k$:
For IF neuron, $\forall t \in [1, LT]$, we will have:

$$\boldsymbol{v}_{IF}^k(t) - \boldsymbol{v}_{IF}^k(t-1) = \mathbf{I}^k\left(\left\lceil \frac{t}{L} \right\rceil\right)/L - \boldsymbol{s}_{IF}^k(t)\theta^k. \tag{24}$$

By accumulating and eliminating redundant terms, we obtain:

$$\boldsymbol{v}_{IF}^k(Lt) - \boldsymbol{v}_{IF}^k(L(t-1)) = \mathbf{I}^k(t) - \sum_{i=L(t-1)+1}^{Lt} \boldsymbol{s}_{IF}^k(t)\theta^k. \tag{25}$$

For IS neuron, $\forall t \in [1, T]$, we will have:

$$\boldsymbol{v}^k(t) - \boldsymbol{v}^k(t-1) = \mathbf{I}^k(t) - \boldsymbol{s}^k(t)\theta^k + \alpha^k(t)\theta^k$$
$$= \mathbf{I}^k(t) - \hat{\boldsymbol{s}}^k(t)\theta^k. \tag{26}$$

Then, combine eq.(25) and eq.(26):

$$\sum_{i=L(t-1)+1}^{Lt} \boldsymbol{s}_{IF}^k(t)\theta^k - \hat{\boldsymbol{s}}^k(t)\theta^k$$
$$= \boldsymbol{v}^k(t) - \boldsymbol{v}^k(t-1) - \left(\boldsymbol{v}_{IF}^k(Lt) - \boldsymbol{v}_{IF}^k(L(t-1))\right).$$
$$= \boldsymbol{v}^k(t) - \boldsymbol{v}_{IF}^k(Lt) \tag{27}$$

Since $\boldsymbol{v}_{IF}^k(L(t-1)) = \boldsymbol{v}^k(t-1) \in [0, \theta^k)$, according to Lemma 1, we will have $\boldsymbol{v}^k(t) \in [0, \theta^k)$ and $\boldsymbol{v}_{IF}^k(Lt) \in [0, \theta^k)$, and then:

$$\left| \sum_{i=L(t-1)+1}^{Lt} \boldsymbol{s}_{IF}^k(t)\theta^k - \hat{\boldsymbol{s}}^k(t)\theta^k \right|$$

$$= \left| \boldsymbol{v}^k(t) - \boldsymbol{v}_{IF}^k(Lt) \right| \in [0, \theta^k). \tag{28}$$

Since $\sum_{i=L(t-1)+1}^{Lt} \boldsymbol{s}_{IF}^k(t)$ and $\hat{\boldsymbol{s}}^k(t)$ are integers:

$$\left| \sum_{i=L(t-1)+1}^{Lt} \boldsymbol{s}_{IF}^k(t)\theta^k - \hat{\boldsymbol{s}}^k(t)\theta^k \right| = 0. \tag{29}$$

Finally, correcting the time step of the IF neuron yields:

$$\hat{\boldsymbol{s}}^k(t) = \sum_{i=1}^{L} \boldsymbol{s}_{IF}^k(i). \tag{30}$$

$\square$

In addition, according to our Theorem 2 and Remark 1, we have $\boldsymbol{\alpha} = 2^{n-j-1}$, which can also be represented by the IF neuron, i.e., $\boldsymbol{\alpha}(t) = \sum_{i=1}^{L'} \boldsymbol{s}_{IF}^k(i)$. Therefore, the output of our IS neuron can be represented by multiple IF neurons.

# G  CONVERT NONLINEAR OPERATION IN LLM

The core principle of our framework is to ensure that the cumulative outputs across the spiking time window closely approximate the corresponding outputs of the quantized ANN. As a concrete example, we illustrate the conversion process using LLaMA-2 Touvron et al. (2023), as shown in Figure, along with its corresponding architecture in Figure. This demonstrates the construction of its spiking counterpart, referred to as Spiking LLaMA. Further details are provided below.

**Attention Layer.** The attention architecture of our method is presented as follows:

$$\mathbf{Q}_q \approx \sum_{t=1}^{T} \mathbf{Q}_{s,t}, \; \mathbf{K}_q \approx \sum_{t=1}^{T} \mathbf{K}_{s,t}, \; \mathbf{V}_q \approx \sum_{t=1}^{T} \mathbf{V}_{s,t}, \tag{31}$$

where $\mathbf{Q}_q$, $\mathbf{K}_q$ and $\mathbf{V}_q$ denote the quantized query $\mathbf{Q}$, key $\mathbf{K}$ and value $\mathbf{V}$, and $\mathbf{Q}_{s,t}$, $\mathbf{K}_{s,t}$ and $\mathbf{V}_{s,t}$ denote the spiking query $\mathbf{Q}$, key $\mathbf{K}$ and value $\mathbf{V}$ at time step $t$.

Then, we need to ensure the equivalence in Activation-Activation (AA) multiplication in attention, which occurs between the query $\mathbf{Q}$ and key $\mathbf{K}$ as well as attention array $\mathbf{A} = \mathbf{QK}$ and value $\mathbf{V}$. Fortunately, You et al., You et al. (2024) have provided the equivalence operation for such AA multiplication. Specifically, taking the multiplication between query and key as an example, it can be written as:

$$\mathbf{A}_q = \mathbf{Q}_q \cdot \mathbf{K}_q \approx \sum_{t=1}^{T} \mathbf{Q}_{s,t} \cdot \sum_{t=1}^{T} \mathbf{K}_{s,t}$$

$$= \sum_{t=1}^{T} (\mathbf{S}_{Q,t} \cdot \mathbf{K}_{s,t} + \mathbf{Q}_{s,t} \cdot \mathbf{S}_{K,t} - \mathbf{Q}_{s,t} \cdot \mathbf{K}_{s,t}) \tag{32}$$

where $\mathbf{S}_{Q,t}$ and $\mathbf{S}_{K,t}$ represent the accumulated spike output of query and key from 0 to $t$. Therefore, the result of AA multiplication at each time $t$ is $\mathbf{S}_{Q,t} \cdot \mathbf{K}_{s,t} + \mathbf{Q}_{s,t} \cdot \mathbf{S}_{K,t} - \mathbf{Q}_{s,t} \cdot \mathbf{K}_{s,t}$.

**Spiking Softmax, Spiking LayerNorm and Spiking SiLU Activation.** Inspired by You et al. (2024), we use the following process to enable Softmax, Layernorm, and SiLU activation in SNN.

$$\mathbf{X}(t) = \mathbf{X}(t-1) + x(t), \tag{33}$$

$$\mathbf{O}(t) = \phi(\mathbf{X}(t)), \tag{34}$$

$$o(t) = \mathbf{O}(t) - \mathbf{O}(t-1), \tag{35}$$

where $\mathbf{X}(t)$ is the accumulated input at $t$ time-step; $x(t)$ is the input at $t$ time-step; $\phi(\cdot)$ is the Softmax, Layernorm and SiLU activation and $o(t)$ is the output at $t$ time step. By using this method, the accumulated outputs of these operations are equivalent to those in a quantized ANN.

**Spiking MLP.** As a pivotal component within the SpikingLLM, the design of the spiking MLP also needs to ensure that the accumulated output remains equivalent to the quantized MLP. Except for the linear layers in MLP, the most important operation is the Activation-Activation (AA) Hadamard product, which exists between the output of a linear layer and the output of the spiking SiLU function. The AA Hadamard product can be written as

$$\mathbf{A} \odot \mathbf{B} = \sum_{t=1}^{T} \mathbf{A}_t \odot \sum_{t=1}^{T} \mathbf{B}_t \tag{36}$$

$$= \sum_{t=1}^{T} \left( \mathbf{A}_t \odot \mathbf{B}_t + \sum_{i=1, i \neq m}^{T} \frac{\mathbf{A}_t \odot \mathbf{B}_i + \mathbf{A}_i \odot \mathbf{B}_t}{2} \right),$$

where $\mathbf{A}$ denotes the output of the **up_proj** and $\mathbf{B}$ denotes the output of Spiking SiLU. Therefore, the result of Hadamard product at each time $t$ is $\mathbf{A}_t \odot \mathbf{B}_t + \sum_{i=1, i \neq m}^{T} \frac{\mathbf{A}_t \odot \mathbf{B}_i + \mathbf{A}_i \odot \mathbf{B}_t}{2}$.

