# OpenReview forum: "How to Get Spiking LLMs? A Dual ANN-to-SNN Conversion with Layer-Wise Calibration"
_ICLR.cc/2026/Conference — Submitted to ICLR 2026_

### Official Review · Reviewer_jzES · 2025-10-27

**Soundness:** 2
**Presentation:** 1
**Contribution:** 1
**Rating:** 2
**Confidence:** 5

**Summary:**

This paper presents an ANN-to-SNN conversion framework for spiking LLMs. It employs the Integer Spiking (IS) neuron, which has multiple thresholds and fires multi-bit spikes. Building on the IS neuron, Quantized ANN (QANN) can be converted to SNN without training another tailored ANN. The proposed conversion method also utilizes the layer-wise calibration to enhance the performance.

**Strengths:**

1. Theoretical analyses of the error bounds of conversion prove the effectiveness of the layer-wise calibration.
2. Compared to conventional ANN2SNN conversion approaches, the proposed framework does not require training a tailored ANN, which is important for LLM conversion.

**Weaknesses:**

1. The contributions of this paper are incremental. First, previous work [1] has already proposed layer-wise calibration methods. This paper hides this technical background and avoids citing the prior research. Furthermore, the multi-bit spike neuron (i.e., M-HT neuron [2], Burst spikes [3], etc.) has also been proposed. Additionally, there is also research [4] that converts QANNs to SNNs. I believe the contributions of this paper are limited to using multi-bit spike neurons to convert QANNs to SNNs.
2. Critical technical details are missing. This paper does not describe how key components of the Transformer, including Layer Norm, Softmax, and matrix multiplication, are converted to a form suitable for SNNs.
3. Lack of energy efficiency analysis. This paper does not analyze the energy efficiency advantages of the converted SNN over the original QANN. Furthermore, the proposed method employs multi-bit spikes. While this facilitates the conversion of QANN to SNN, it also introduces additional computational overhead in the linear layer. This paper does not analyze this impact either.
4. This paper lacks comparisons with state-of-the-art conversion methods, such as SpikeZIP-TF [5], in terms of performance and energy efficiency.



[1] Li, Yuhang, et al. "A free lunch from ANN: Towards efficient, accurate spiking neural networks calibration." *International conference on machine learning*. PMLR, 2021.

[2] Hao, Zecheng, et al. "LM-HT SNN: Enhancing the performance of SNN to ANN counterpart through learnable multi-hierarchical threshold model." *Advances in Neural Information Processing Systems* 37 (2024): 101905-101927.

[3] Li, Yang, and Yi Zeng. "Efficient and Accurate Conversion of Spiking Neural Network with Burst Spikes." *Proceedings of the Thirty-First International Joint Conference on Artificial Intelligence*. 2022.

[4] Yang, Yuchen, et al. "NeuBridge: bridging quantized activations and spiking neurons for ANN-SNN conversion." *Neuromorphic Computing and Engineering* 5.2 (2025): 024018.

[5] You, Kang, et al. "SpikeZIP-TF: Conversion is All You Need for Transformer-based SNN." *International Conference on Machine Learning*. PMLR, 2024.

**Questions:**

See weaknesses

---

### Official Review · Reviewer_VqZM · 2025-10-31

**Soundness:** 1
**Presentation:** 3
**Contribution:** 2
**Rating:** 4
**Confidence:** 4

**Summary:**

This paper presents a two-stage ANN-to-SNN conversion framework that improves scalability to large language models. The authors propose a layer-wise calibration method that significantly reduces conversion errors. The framework is demonstrated through converting pre-trained LLaMA models into spiking large language models (SNN-LLMs), achieving performance comparable to state-of-the-art quantization methods.

**Strengths:**

- The paper explores ANN-to-SNN conversion in the context of very large models.
- The proposed layer-wise calibration approach is theoretically motivated and empirically validated.

**Weaknesses:**

- The paper claims to be “training-free,” but the calibration step still appears to involve optimization via BPTT or similar gradient-based tuning of neuronal thresholds. This step could reintroduce significant computational costs, especially for LLM-scale models.
- There is a mismatch between the stated motivation and the evaluation metrics. The central motivation of ANN-to-SNN conversion is energy efficiency and neuromorphic deployability, yet the experiments focus solely on accuracy and perplexity benchmarks. The paper would benefit from including energy, latency, or hardware feasibility analyses.
- The proposed models (tested on LLaMA-2 and LLaMA-3) are far beyond the capacity of existing neuromorphic chips (e.g., Loihi 2, TrueNorth). The authors should clarify which hardware platforms are envisioned for deployment, and whether the approach offers any practical energy advantage over quantized or sparsified ANN-based LLMs.
- The paper employs graded spikes. This design choice should be explicitly stated early in the paper and discussed in terms of hardware constraints, as graded spikes may complicate neuromorphic implementation.

**Questions:**

See in Weaknesses

---

### Official Review · Reviewer_pCuY · 2025-11-01

**Soundness:** 3
**Presentation:** 3
**Contribution:** 3
**Rating:** 6
**Confidence:** 5

**Summary:**

This paper proposes a novel "dual ANN-to-SNN conversion" framework to address the challenge of efficiently converting Large Language Models into Spiking Neural Networks. The core idea is to introduce a Quantized ANN as an intermediate model, decoupling the conversion process into two stages: ANN→QANN and QANN→SNN. To achieve precise modeling, the authors design a new Integrate-and-Spike with Reset neuron and provide a rigorous theoretical proof of its functional equivalence to standard Integrate-and-Fire neurons, which lays a solid foundation for hardware implementation. The paper also conducts theoretical error decomposition and proposes a layer-wise calibration method to optimize performance. Experiments on LLaMA models demonstrate the effectiveness of the approach.

**Strengths:**

1.Decoupling the ANN-to-SNN conversion into "quantization" and "temporal dynamicization" sub-problems via a QANN intermediate bridge is a ingenious approach.

2.The proposed IS neuron model and its equivalence proof to IF neurons are proposed.

**Weaknesses:**

1.The method heavily relies on the discrete integer outputs of the QANN. For modern LLMs using non-uniform quantization or containing complex activation functions (e.g., SwiGLU), their discretized values may be difficult to precisely reconstruct with a finite number of spike timesteps, potentially limiting the method's generalizability.

2.Although the equivalence between IS and IF neurons is proven, this implementation requires a large number of IF neurons and timesteps to simulate a single IS neuron. The paper completely ignores the area overhead (requiring more neurons) and time overhead (requiring more timesteps) introduced by this mapping, which is crucial for evaluating the actual energy efficiency of the SNN.

**Questions:**

As in weakness.

---

### Official Review · Reviewer_S5K6 · 2025-11-07

**Soundness:** 3
**Presentation:** 3
**Contribution:** 3
**Rating:** 4
**Confidence:** 4

**Summary:**

This paper proposes a novel method for efficiently converting LLMs into SNNs.. The core methodology includes: (1) introducing a membrane potential alignment mechanism during ANN-to-SNN conversion to reduce activation distribution mismatches; (2) adopting a parameter-efficient fine-tuning strategy that only learns neuronal firing thresholds and initial membrane potentials per layer, while keeping pretrained weights frozen; and (3) further incorporating grouped activation scaling, where a small number of learnable parameters are shared across activation groups to compensate for quantization and clipping errors. Experiments on LLaMA show significant improvements over conventional SNN conversion baselines, achieving competitive zero-shot accuracy and language modeling performance.

**Strengths:**

1.The paper goes beyond empirical heuristics by providing formal analysis, which bounds the approximation error between SNN outputs and ANN activations within a finite number of timesteps.

2.The proposed membrane potential alignment and grouped activation fine-tuning are theoretically grounded and achieve a good trade-off between parameter efficiency and performance.

**Weaknesses:**

1.Despite claims about low-power deployment, all experiments are simulation-based. No energy, latency, or throughput measurements are reported on neuromorphic hardware such as Loihi, making it impossible to validate the practical feasibility of edge deployment.

2.The evaluation is only on LLaMA-2 and LLaMA-3, with no testing on other mainstream LLM architectures.

**Questions:**

1.The IS neurons produce only non-negative and bounded activation. How can they effectively emulate SwiGLU, which involves negative activation values?

2.Although the paper establishes functional equivalence between IS and IF neurons, the practical implementation requires multiple IF neurons and a large number of timesteps to emulate a single IS neuron. Have the authors considered the resulting area overhead due to increased neuron count? How do these factors impact the actual energy efficiency and throughput when deployed on real neuromorphic hardware?

---

### Meta-Review · Area_Chair_PLRc · 2025-12-19

**Summary:**

The paper introduces a method for converting Large Language Models (LLMs) into Spiking Neural Networks (SNNs). It proposes several innovations, including a membrane potential alignment mechanism, a parameter-efficient fine-tuning strategy, and grouped activation scaling. The authors demonstrate the effectiveness of these methods on LLaMA models, showing improvements in accuracy and language modeling performance.

**Reviewer Concerns:**

Reviewer S5K6: The lack of energy, latency, or throughput measurements on neuromorphic hardware like Loihi makes it impossible to evaluate the practical feasibility of edge deployment. Furthermore, there is no testing on other mainstream LLM architectures beyond LLaMA models. The concerns about the feasibility of implementing SwiGLU and the area and time overheads introduced by the increased neuron count remain unresolved.

Reviewer pCuY: The reliance on QANN outputs, which may not accurately reconstruct activation values for modern LLMs using complex activation functions, limits the generalizability of the method. The question about the area and time overheads due to simulating IS neurons with IF neurons remains unaddressed.

Reviewer VqZM: The calibration step may still involve optimization, potentially reintroducing computational costs. The lack of energy efficiency analysis and the mismatch between the motivation (neuromorphic deployability) and the evaluation metrics (accuracy/perplexity) are significant concerns. Furthermore, there is no clarification on the hardware platforms for deployment.

Reviewer jzES: The contributions are deemed incremental, with prior work already addressing similar methods. Missing technical details, such as the conversion of key Transformer components (Layer Norm, Softmax), and the lack of an energy efficiency analysis or comparison with state-of-the-art methods (like SpikeZIP-TF) are significant drawbacks.

**Reviewer Scores:**

Since there is no response for this paper, I believe the reviewers would maintain their original scores.

---

### Decision · Program_Chairs · 2026-01-26

Reject